# Recovery of neuropsychological function following abstinence from alcohol in adults diagnosed with an alcohol use disorder: Protocol for a systematic review of longitudinal studies

Anna Powell [1,2]☯*, Harry Sumnall[2,3]☯, Jessica Smith[2,3]‡, Rebecca Kuiper[1,2]‡, Catharine Montgomery [1,2]☯

1 Faculty of Health, School of Psychology, Liverpool John Moores University, Liverpool, United Kingdom, 2 Liverpool Centre for Alcohol Research, University of Liverpool, Liverpool, United Kingdom, 3 Faculty of Health, Public Health Institute, Liverpool John Moores University, Liverpool, United Kingdom

☯ These authors contributed equally to this work.
‡ These authors also contributed equally to this work.
* a.powell@2019.ljmu.ac.uk

**Data Availability Statement:** No datasets were generated or analysed during the current study. All

## Abstract

### Background

Alcohol use disorders (AUD) associate with structural and functional brain differences, including impairments in neuropsychological functions; however, review level research (largely cross-sectional) is inconsistent with regards to recovery of such functions following abstinence. Such recovery is important, as these impairments associate with treatment outcomes and quality of life.

### Objective(s)

To assess neuropsychological function recovery following abstinence in individuals with a clinical AUD diagnosis. The secondary objective is to assess predictors of neuropsychological recovery in AUD.

### Methods

Four electronic databases (APA PsycInfo, EBSCO MEDLINE, CINAHL, Web of Science Core Collection) will be searched between 1999–2022, with search strategies adapted for each source. Study reporting will follow the Joanna Briggs Institute (JBI) Manual for Evidence Synthesis, study quality will be assessed using the JBI Checklist for Cohort Studies. Eligible studies are those with a longitudinal design that assessed neuropsychological recovery following abstinence from alcohol in adults with a clinical diagnosis of AUD. Studies will be excluded if participant group is defined by another or co-morbid condition/injury, or by relapse.

relevant data from this study will be made available upon study completion.

**Funding:** The author(s) received no specific funding for this work.

**Competing interests:** The authors have declared that no competing interests exist.

## Results

This is an ongoing review. As of July 2022, the review protocol is registered on PROSPERO (CRD42022308686), searches have been conducted, and screening is in progress. Results are predicted to be complete by October 2022.

## Conclusions

Comparing data on neuropsychological recovery from AUD will improve understanding of the impact of alcohol on the brain, and the relationship between AUD recovery and quality of life/treatment outcomes. It may provide information that could one day inform aspects of treatment and aftercare (e.g., options for cognitive training of functions that do not improve on their own).

## Introduction

Globally, alcohol is the seventh leading risk for death and disability, with all-cause mortality risk rising with consumption [1]. Adult per-capita alcohol consumption has been increasing since 1990, and trends are predicted to rise until 2030 [2]. Furthermore, 5.1% of all individuals aged 15+ are estimated to have an alcohol use disorder (AUD), though this differs by WHO region (European (8.8%), Americas (8.2%), Western Pacific (4.7%), African (3.7), Eastern Mediterranean (0.8%)) [3]. AUD describes continued alcohol use despite negative consequences [4,5]. Prolonged use can be neurotoxic, possibly via neuronal loss through disrupting neurogenesis, oxidative stress, or glutamate excitotoxicity [6]. Thiamine deficiency causes indirect damage [7]. A diagnosis occurring more in AUD than the general population is alcohol-related brain injury (ARBI), affecting an estimated 35%, though not all will be diagnosed [8]. ARBI is an umbrella term for major neurocognitive disorders caused by drinking [9]. There is a lack of consensus on which conditions are ARBI, though it generally includes Wernicke's encephalopathy, Korsakoff's Syndrome (usually preceded by Wernicke's [10], together Wernicke-Korsakoff's Syndrome), and alcohol related dementia [9].

While not everyone with an AUD is diagnosed with an ARBI, there is review level research linking uncomplicated AUD with brain differences, though this seems more pronounced in diagnosed ARBI [11]. Brain differences in AUD occur across structure and function, including within neurotransmitter and metabolic systems [11], grey and white matter [11–16], and event-related potential markers of attentional capacity [17].

Furthermore, a variety of neuropsychological functions are impaired in AUD, including inhibition, set-shifting, working memory, problem solving, planning, attention, reasoning/abstraction, processing speed, visuospatial abilities, verbal memory, verbal learning, verbal fluency, visual memory, visual learning, intelligence [18–20]. Other deficits include social cognition, such as Theory of Mind [21,22], and facial emotion recognition [21,23]. The severity of the latter associates with alcohol use duration and depressive symptoms [21]. Fauth-Bühler and Kiefer [24] found reduced brain response to emotional stimuli (particularly in limbic regions).

Consequently, AUD is associated with multiple neuropsychological impairments (though most of this literature is cross-sectional, so cannot exclude pre-existing differences), it is important to understand whether these can recover with abstinence. A prospective review [25] consistent improved sustained attention, but inconsistencies for attention, memory, working

memory, executive functions, and processing speed. Poorer baseline performance, number of detoxifications, family history, and smoking were all moderating factors for neurocognitive recovery.

Two methodologically similar meta-analyses across varying levels of abstinence by Stavro, Pelletier [19] and Crowe [18], found conflicting results. While both indicated impairment across all functions tested (except IQ in Stavro, Pelletier [19]), one found recovery of all domains (inhibition not included as too few papers) by a year of abstinence [19], while Crowe [18] found a wide variety of persisting impairments at all three time periods, including after a year (particularly visual/verbal memory, executive functioning, processing speed, and verbal learning, and except working memory).

Therefore, while there is support for recovery of neuropsychological functions with abstinence, evidence is inconsistent, and there are methodological issues. Firstly, the studies included in Crowe [18], Stavro, Pelletier [19] were largely cross-sectional, limiting conclusions about causality [26]. Secondly, the most suitable review is Schulte, Cousijn [25], as it included only longitudinal studies with controls (with many papers having tested controls at least twice, reducing impact of AUD practice effects), however this still found inconsistent results.

The proposed systematic review specifically aims to investigate recovery of neuropsychological function following abstinence in AUD, addressing the limitations discussed above. This research is important, because a) functional impairments in AUD can reduce a person's quality of life [27], and b) these impairments are linked to treatment outcomes [28], so how they recover may inform methods to support individuals through AUD recovery.

## Objective(s)

To assess neuropsychological function recovery following abstinence in individuals with a clinical AUD diagnosis. The secondary objective is to assess predictors of neuropsychological recovery in AUD.

## Methods

### Protocol

The protocol used the Joanna Briggs Institute (JBI; [29]) Manual for Evidence Synthesis, and PRESS [30]/ PRISMA-P checklists ([31]).

## Eligibility criteria

### Population

Adults with a clinical diagnosis of AUD and in recovery (abstinent at least two weeks [25]) for at least the first recovery time point. Overall mean age shall be 18–64 years at baseline, as alcohol use, related risk, and brain structure/function change across lifespan, but this is likely most pronounced in young people (aged < 18) and older adults (aged >64) thus reducing comparability [32–35]. It is likely that many people (indeed likely the majority) in a clinical sample being treated for AUD will also use other substances [36,37], therefore if participants are reported as consuming other substances, to be included, a study cannot be defined by this and alcohol must be the primary (a study will not be included if it specifically recruits individuals with AUD who also use other substances). An SUD that is particularly highly comorbid with AUD is tobacco use disorders [38,39], and therefore if a study reports some participants as having a comorbid tobacco use disorder (but does not specifically recruit individuals with AUD who use tobacco), then it can be included. If a study includes groups of individuals with

different types of SUD including AUD, it can be included so long as the study clearly reports AUD subgroup results.

**Exposure.** Abstinence from alcohol in recovery from an AUD, defined as either a clinical diagnosis of AUD (mild, moderate, or severe) as per DSM-5 (2013), alcohol dependence/abuse as per DSM-IV (1994), or alcohol dependence/harmful use, as per ICD-10 (1994) or ICD-11 (2019), for diagnostic consistency.

## Comparators

i) adults without AUD; ii) adults with a different severity of AUD; iii) abstinence duration assessed by regression (including analysis of variance), as in Schulte, Cousijn [25].

**Outcome.** Primary outcome is change in neuropsychological function from baseline (which may occur before/during active AUD, or in early recovery) to last available follow-up. This must have been assessed at least twice using a validated self-report/task measure or analogous measure, or as clinical diagnoses/progression of neuropsychological impairment.

**Study design.** Longitudinal (cohort: prospective or retrospective), published since the year 1999 to account for the introduction of various contemporary neuroscientific theories of addiction [40], such as [41–43].

**Exclusion criteria.** Grey literature; animal studies; studies not published in English (as this is an unfunded review, though these shall be described and excluded at the full-text stage; Centre for Reviews and Dissemination [44], with language listed as reason for exclusion); population defined by another or co-morbid condition (such as a major psychiatric condition, head trauma, ARBI diagnosis, or co-morbid or secondary other substance use disorder, or alcohol relapse).

## Search strategy

A four-stage search strategy will be used: 1) an initial search of databases (CINAHL, APA PsycInfo, EBSCO MEDLINE, Web of Science Core Collection) using pre-specified keywords (alcohol dependence, alcohol use disorder, cognitive function) has identified other keywords and subject headings, to be followed by: 2) full strategy searching across all sources, 3) hand-searching reference lists of included papers, 4) forward searching, with articles citing included studies screened for relevance. Search filters will be used where possible. Clinical trials registers will not be searched, as these are likely to bring up papers on intervention efficacy, rather than neuropsychological assessment/recovery. The study list will be circulated amongst all authors to enable identification of any missing studies. Searches will be re-run prior to final analysis. See S1 File for search strategies for each source.

## Data management and selection process

Search strategy results (references, abstracts, and full texts where available) will be transferred into EndNote, for storage and grouping by decision. Pre-screening exclusion (e.g., duplicates identified by Endnote, or records removed via search source filters such as participant age/species or publication date) shall be documented. Papers will be screened (first via titles/abstracts) using review criteria. Initial screening will be against two preliminary criteria: a) study participants are human adults aged 18+, and b) study appears to longitudinally assess recovery of neuropsychological function from AUD.

When studies meet above initial criteria, an attempt will be made to obtain full texts and key information for full criteria screening, and data extraction. If necessary, full texts will be obtained via inter-library loan, and/or contacting authors. If key information is not received within a month of contact, the text will be excluded. Rationale for exclusion at this full-text

screening stage shall be documented in a table. Screening shall be conducted independently by three assessors, one of whom (AP) will screen all data, and the other two (JS & RK) shall each screen half, for fidelity. Any uncertainties shall be discussed between the research team. Inter-reviewer consistency shall be determined prior to screening, by the three assessors all screening 25 randomly selected sources and establishing a kappa statistic.

Duplicates will be identified, including identical records and papers describing different outcomes or time-points of the same study. Identifiers will be used, including paper and author name, description of methods, participant numbers, baseline data, study dates/durations. If necessary, authors will be contacted. If multiple articles describe the same study, a primary paper will be chosen as the main source of results. This shall be decided via discussion between reviewers. Papers reporting different relevant outcomes but not chosen as the primary paper will be considered secondary sources of study information. Management of the selection process will be supported via EndNote and Microsoft Excel. Study selection will be presented in a PRISMA 2020 flow diagram.

## Data extraction

A data extraction from based on the JBI manual has been created (S2 File), including definitions of each element for consistency. Data extraction will be undertaken by AP, and the spreadsheet shared with other team members, so 10% of articles can be checked for fidelity. The following details will be extracted: authors; title; year; funding; conflicts of interest; design; setting; location; participant characteristics (age, sex, gender, sample size, exact diagnosis, diagnosis length, age of onset, no. treatment attempts, comorbidities, substance use, details of comparison groups, attrition details); recruitment/follow-up procedures; data relating to change in neuropsychological function (measurement, analysis, results, statistical significance, and confound adjustments); data relating to secondary aims (characteristics reported as predictors of neuropsychological recovery) including measurement and results.

Data extraction and quality appraisal will be piloted by AP on a sample of five full-text papers (selected for wide-ranging outcome measures and time-points). This method will inform refinement of data extraction and quality appraisal [44].

## Quality assessment

The JBI Checklist for Cohort studies [29] will be applied at the primary outcome level to provide appraisal of study methods, risk of bias, and validity of results. Scoring is rated as 'yes', 'no', 'unclear' or 'not applicable'. Responses of 'yes' (1) will be summed against the maximum total (11) and scores transformed into percentages and ratings (poor = 49%, moderate = 50–69%, good = 70% onwards), as in Hall, Le [45]. Scores will not be used to exclude studies [44] but displayed in a table to inform appraisal. At least 10% of this screening will be independently conducted for accuracy.

## Data synthesis

Due to the heterogenous nature of methodologies, a narrative synthesis will be produced, and meta-bias shall not be assessed. Popay, Roberts [46] and the University of York's Centre for Reviews and Dissemination [44] suggest four key elements of a narrative synthesis; 1) developing a theory of how the intervention works, 2) developing a preliminary synthesis of results, 3) exploring relationships in the data, 4) assessing robustness of the synthesis. Our review will not be evaluating an intervention, therefore as in Heirene, Roderique-Davies [47] we will not use the first feature.

The synthesis will group, describe and discuss data according to functions assessed, and neuropsychological measures used, using Lezak, Howieson [48] for guidance. Some studies may be represented multiple times. Key study aspects will be summarised within groups, and then differences/similarities will be compared to draw conclusions, with regards to review outcomes.

Tables and figures will be used to support the synthesis, including a table of study characteristics, and a table summarising the measures used in each study, domains assessed, and outcomes. Both tables shall be grouped by risk of bias, as suggested by Cochrane Handbook Chapter 12.4.1. We also aim to provide a review matrix mapping recovery of neuropsychological function, in a similar fashion to the matrix created by Pask, Dell'Olio [49] of opiate impacts on cognition. Finally, robustness of findings will be discussed using JBI Quality Appraisal Checklist results and limitations of the synthesis process itself.

## Amendments

If amendments are made to the methodology outlined here, they will be recorded along with rationale and date. AP shall be responsible for documenting this, but any changes will be approved by all authors. Changes will not be incorporated into the protocol, but will be added to the PROSPERO registration, and will be summarised in the final manuscript.

## Supporting information

**S1 Checklist. PRISMA-P 2015 checklist.**
(PDF)

**S1 File. Systematic search strategies for APA PsycInfo, EBSCO MEDLINE, CINAHL, and Web of Science.**
(PDF)

**S2 File. Data extraction form.**
(PDF)

## Author Contributions

**Conceptualization:** Anna Powell, Harry Sumnall, Catharine Montgomery.

**Methodology:** Anna Powell, Harry Sumnall, Catharine Montgomery.

**Project administration:** Anna Powell.

**Supervision:** Harry Sumnall, Catharine Montgomery.

**Writing – original draft:** Anna Powell.

**Writing – review & editing:** Harry Sumnall, Jessica Smith, Rebecca Kuiper, Catharine Montgomery.

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
