## [Decision Letter · Decision Letter 0]

23 Jun 2022

PONE-D-22-05405

Recovery of neuropsychological function following abstinence from alcohol in adults diagnosed with an Alcohol Use Disorder: Protocol for a systematic review of longitudinal studies

PLOS ONE

Dear Dr. Powell,

Thank you for submitting your manuscript to PLOS ONE. After careful consideration, we feel that it has merit but does not fully meet PLOS ONE’s publication criteria as it currently stands. Therefore, we invite you to submit a revised version of the manuscript that addresses the points raised during the review process.

We look forward to receiving your revised manuscript.

Kind regards,

Matthew J. Gullo

Academic Editor

PLOS ONE

2. Please note that in order to use the direct billing option the corresponding author must be affiliated with the chosen institute. Please either amend your manuscript to change the affiliation or corresponding author, or email us at plosone@plos.org with a request to remove this option.

3. We note that this manuscript is a systematic review or meta-analysis; our author guidelines therefore require that you use PRISMA guidance to help improve reporting quality of this type of study. Please upload copies of the completed PRISMA checklist as Supporting Information with a file name “PRISMA checklist”.

Reviewers' comments:

Reviewer's Responses to Questions

**Comments to the Author**

1. Does the manuscript provide a valid rationale for the proposed study, with clearly identified and justified research questions?

Reviewer #1: Yes

2. Is the protocol technically sound and planned in a manner that will lead to a meaningful outcome and allow testing the stated hypotheses?

Reviewer #1: Partly

3. Is the methodology feasible and described in sufficient detail to allow the work to be replicable?

Reviewer #1: No

4. Have the authors described where all data underlying the findings will be made available when the study is complete?

Reviewer #1: No

5. Is the manuscript presented in an intelligible fashion and written in standard English?

Reviewer #1: Yes

6. Review Comments to the Author

You may also provide optional suggestions and comments to authors that they might find helpful in planning their study.

Reviewer #1: Powell and colleagues describe the protocol for a systematic review to examine changes in neuropsychological functioning in patients with alcohol use disorder. The need for a systematic review in this field is briefly justified, but could be made more detailed. Existing evidence from cross sectional studies is included quite loosely in the introduction. Findings on changes in brain function and structure are addressed very superficially. All in all, the introduction could be a bit more straightforward.

The protocol is well described and largely follows current guidelines for systematic reviews. The literature databases to be searched, the search strings as well as the inclusion and exclusion criteria are mentioned. The data extraction and quality control follow a defined procedure.

However, there are also some concerns that should be addressed before publication.

1) Why are only publications after the year 2000 included? The stated reason "to account for the introduction of various contemporary neuroscientific theories of addiction" is not comprehensible to me. I also could not find an answer to this question in the mentioned reference. Please justify this limitation in detail and give appropriate references.

2) The review should include studies whose sample has an age range of 18-64 years, which seems reasonable because of the changes in cognitive performance. However, it is also stated that: "Studies can be included if =60% are aged 18-64". This is difficult for me to understand. Is there any meaningful rationale for this threshold? If not, I would recommend excluding these studies to avoid bias in interpretation.

3) The next point is quite similar: "If participants are reported as consuming other substances, alcohol must be the primary and the study cannot be defined by this". On the one hand, it is unclear what is meant by "the study cannot be defined by this". Second, different psychotropic substances have different pharmacological profiles and can affect cognitive functions quite differently. If the effects of alcohol are to be investigated, then it would make sense to limit the included studies to AUD. Otherwise, there is also a risk of significant interpretation bias. Because of the high comorbidity of tobacco use disorders, these would be the only SUD that would be acceptable.

7. PLOS authors have the option to publish the peer review history of their article (what does this mean?). If published, this will include your full peer review and any attached files.

Reviewer #1: No

---

## [Author Response · Author response to Decision Letter 0]

2 Aug 2022

Manuscript PONE-D-22-05405

Response to Reviewers

Dear Professor Matthew Gullo,

Thank you for giving us the opportunity to submit a revised draft of our manuscript titled Recovery of neuropsychological function following abstinence from alcohol in adults diagnosed with an Alcohol Use Disorder: Protocol for a systematic review of longitudinal studies, to PLOS ONE. We appreciate the time and effort that you and the reviewer have dedicated to providing valuable feedback on our manuscript. We are grateful for the insightful comments on our paper. We have been able to incorporate changes that reflect most of the suggestions provided, which are highlighted within the manuscript. Please see below for a point-by-point response to the reviewer. All page and line numbers refer to the revised manuscript file with tracked changes. 

Reviewer: 1

1. Why are only publications after the year 2000 included? The stated reason "to account for the introduction of various contemporary neuroscientific theories of addiction" is not comprehensible to me. I also could not find an answer to this question in the mentioned reference. Please justify this limitation in detail and give appropriate references.

Response: Thanks for this comment, to clarify, the reference given (Fernández-Serrano et al., 2011) refers to this on page 5 of 30 (line 3) when viewed as a PDF, listing the following as an inclusion criteria; “Manuscripts published between 1999 and 2009 (including papers ahead of print available at databases before January 2010): this criteria was meant to review only those studies published during the last decade, encompassing the period after the surge of contemporary neuroscientific models of addiction (e.g., Everitt and Robbins, 2005; Goldstein and Volkow, 2002; Koob and Le Moal, 2001) and filtering earlier studies, many of which had important methodological drawbacks (see Verdejo-García et al., 2004 for review).” A cut-off of 2000 was chosen for the current manuscript in combination with this argument, and to give a realistic limit to the number of papers which would require screening. However, we appreciate the confusion, so we have changed the cut-off to publications after the year 1999 (page 2, line 9; page 6, line 7), in keeping with the reference cited, and have cited Everitt and Robbins (2005); Goldstein and Volkow (2002); Koob and Le Moal (2001) on page 6 (lines 8-9) for more information. 

2. The review should include studies whose sample has an age range of 18-64 years, which seems reasonable because of the changes in cognitive performance. However, it is also stated that: "Studies can be included if =60% are aged 18-64". This is difficult for me to understand. Is there any meaningful rationale for this threshold? If not, I would recommend excluding these studies to avoid bias in interpretation.

Response: Thank you for pointing this out, this was an arbitrary threshold set to enable a wider variety of studies to be included. However, we appreciate your comment and wish to avoid interpretation bias, so we have removed this criterion. Therefore, this sentence has been deleted.

3. The next point is quite similar: "If participants are reported as consuming other substances, alcohol must be the primary and the study cannot be defined by this". On the one hand, it is unclear what is meant by "the study cannot be defined by this". Second, different psychotropic substances have different pharmacological profiles and can affect cognitive functions quite differently. If the effects of alcohol are to be investigated, then it would make sense to limit the included studies to AUD. Otherwise, there is also a risk of significant interpretation bias. Because of the high comorbidity of tobacco use disorders, these would be the only SUD that would be acceptable. 

Response: We appreciate the uncertainty as our wording was unclear. We are aware of how prevalent substance use is, indeed in individuals with AUD, Moss et al. (2015) found that only 27.5% used alcohol only (with 32.4% also using tobacco, and 25.3% also using tobacco, cannabis, cocaine and other illicit drugs), while Martin et al. (1996) found that 61% of recruited individuals with AUD reported simultaneous substance use. Indeed, as Moss et al. state, “AD presents with substantial heterogeneity in clinical features, onset age, severity, treatment-seeking, comorbid psychopathology, and non-alcohol substance use” (Moss et al., 2015, p. 2). Therefore, in a similar fashion to how we will include studies where some (but not all) participants have comorbid psychopathology (because it is not realistic to exclude this as comorbidity is so high and not always reported), we wish to do the same with substance use, lest we exclude a significant portion of meaningful data. The phrase “the study cannot be defined by this”, means that to be included, a study cannot specifically recruit individuals with AUD who also use e.g., cocaine. For clarity, the text has been updated on page 5, lines 11-15. 

We agree that due to the high comorbidity of tobacco use disorders, inclusion of studies in which some of the participants are reported as having this comorbidity would be acceptable. The text has been updated to reflect this (page 5, lines 15-17). However, again, if a study population is recruited because of their comorbid AUD and tobacco use disorder (e.g., every single participant is comorbid), this will not be included. 

References

Everitt, B. J., & Robbins, T. W. (2005). Neural systems of reinforcement for drug addiction: from actions to habits to compulsion. Nature neuroscience, 8(11), 1481-1489. 

Fernández-Serrano, M. J., Pérez-García, M., & Verdejo-García, A. (2011). What are the specific vs. generalized effects of drugs of abuse on neuropsychological performance? Neuroscience & Biobehavioral Reviews, 35(3), 377-406. https://doi.org/10.1016/j.neubiorev.2010.04.008

Goldstein, R. Z., & Volkow, N. D. (2002). Drug Addiction and Its Underlying Neurobiological Basis: Neuroimaging Evidence for the Involvement of the Frontal Cortex. American Journal of Psychiatry, 159(10), 1642-1652. https://doi.org/10.1176/appi.ajp.159.10.1642

Koob, G. F., & Le Moal, M. (2001). Drug addiction, dysregulation of reward, and allostasis. Neuropsychopharmacology, 24(2), 97-129. 

Martin, C. S., Clifford, P. R., Maisto, S. A., Earleywine, M., Kirisci, L., & Longabaugh, R. (1996). Polydrug use in an inpatient treatment sample of problem drinkers. Alcoholism, Clinical and Experimental Research, 20(3), 413-417. https://doi.org/10.1111/j.1530-0277.1996.tb01067.x

Moss, H. B., Goldstein, R. B., Chen, C. M., & Yi, H. Y. (2015). Patterns of use of other drugs among those with alcohol dependence: Associations with drinking behavior and psychopathology. Addict Behav, 50, 192-198. https://doi.org/10.1016/j.addbeh.2015.06.041

---

## [Decision Letter · Decision Letter 1]

6 Sep 2022

Recovery of neuropsychological function following abstinence from alcohol in adults diagnosed with an Alcohol Use Disorder: Protocol for a systematic review of longitudinal studies

PONE-D-22-05405R1

Dear Dr. Powell,

We’re pleased to inform you that your manuscript has been judged scientifically suitable for publication and will be formally accepted for publication once it meets all outstanding technical requirements.

Kind regards,

Matthew J. Gullo

Academic Editor

PLOS ONE

Additional Editor Comments (optional):

Reviewers' comments:

Reviewer's Responses to Questions

**Comments to the Author**

1. Does the manuscript provide a valid rationale for the proposed study, with clearly identified and justified research questions?

Reviewer #1: Yes

Reviewer #2: Yes

2. Is the protocol technically sound and planned in a manner that will lead to a meaningful outcome and allow testing the stated hypotheses?

Reviewer #1: Yes

Reviewer #2: Yes

3. Is the methodology feasible and described in sufficient detail to allow the work to be replicable?

Reviewer #1: Yes

Reviewer #2: Yes

4. Have the authors described where all data underlying the findings will be made available when the study is complete?

Reviewer #1: Yes

Reviewer #2: Yes

5. Is the manuscript presented in an intelligible fashion and written in standard English?

Reviewer #1: Yes

Reviewer #2: Yes

6. Review Comments to the Author

You may also provide optional suggestions and comments to authors that they might find helpful in planning their study.

Reviewer #1: My comments have been adequately addressed.

-The manuscript provide a valid rationale for the proposed study, with clearly identified and justified research questions

-The protocol is technically sound

-The methodology is feasible and described in sufficient detail to allow the work to be replicable

-The manuscript is presented in an intelligible fashion and written in standard English

I wish you much success for this work.

Reviewer #2: Thank you for a well-written and clear protocol for this review. The revisions made are clear and it appears you have addressed the issues raised by the first reviewer. The search strategy is reproducible and exhaustive. I look forward to seeing the results of this review.

7. PLOS authors have the option to publish the peer review history of their article (what does this mean?). If published, this will include your full peer review and any attached files.

Reviewer #1: No

Reviewer #2: **Yes: **Amanda Ross-White

---

## [Editor Report · Acceptance letter]

20 Sep 2022

PONE-D-22-05405R1 

Recovery of neuropsychological function following abstinence from alcohol in adults diagnosed with an Alcohol Use Disorder: Protocol for a systematic review of longitudinal studies 

Dear Dr. Powell:

I'm pleased to inform you that your manuscript has been deemed suitable for publication in PLOS ONE. Congratulations! Your manuscript is now with our production department. 

Kind regards, 

on behalf of

Assoc. Prof. Matthew J. Gullo 

Academic Editor

PLOS ONE